# Multilevel Comparison of Indian *Naja* Venoms and Their Cross-Reactivity with Indian Polyvalent Antivenoms

**DOI:** 10.3390/toxins15040258

**Published:** 2023-04-01

**Authors:** Archana Deka, Siddharth Bhatia, Vishal Santra, Omesh K. Bharti, Hmar Tlawmte Lalremsanga, Gerard Martin, Wolfgang Wüster, John B. Owens, Stuart Graham, Robin Doley, Anita Malhotra

**Affiliations:** 1Molecular Toxinology Laboratory, Department of Molecular Biology and Biotechnology, Tezpur University, Tezpur 784028, Assam, India; 2CSIR-Centre for Cellular and Molecular Biology, Laboratory for Conservation of Endangered Species, Hyderabad 500048, Telangana, India; 3Society for Nature Conservation, Research and Community Engagement (CONCERN), Nalikul, Hooghly 712407, West Bengal, India; 4Captive and Field Herpetology, Anglesey LL65 1YU, UK; 5Snake Research Institute, Gujarat Forest Department, Government of Gujarat, Valsad 396050, Gujarat, India; 6State Institute of Health and Family Welfare, Shimla 171009, Himachal Pradesh, India; 7Department of Zoology, Mizoram University, Aizawl 796004, Mizoram, India; 8The Liana Trust, Hunsur 571189, Karnataka, India; 9Molecular Ecology and Evolution @ Bangor (MEEB), School of Natural Sciences, Bangor University, Gwynedd LL57 2UW, UK

**Keywords:** biogeography, biodiversity, venom variation, snakebite, neutralization, antivenomics, *Naja naja*, *Naja kaouthia*, *Naja oxiana*

## Abstract

Snake envenoming is caused by many biological species, rather than a single infectious agent, each with a multiplicity of toxins in their venom. Hence, developing effective treatments is challenging, especially in biodiverse and biogeographically complex countries such as India. The present study represents the first genus-wide proteomics analysis of venom composition across *Naja* species (*N. naja*, *N. oxiana*, and *N. kaouthia*) found in mainland India. Venom proteomes were consistent between individuals from the same localities in terms of the toxin families present, but not in the relative abundance of those in the venom. There appears to be more compositional variation among *N. naja* from different locations than among *N. kaouthia*. Immunoblotting and in vitro neutralization assays indicated cross-reactivity with Indian polyvalent antivenom, in which antibodies raised against *N. naja* are present. However, we observed ineffective neutralization of PLA_2_ activities of *N. naja* venoms from locations distant from the source of immunizing venoms. Antivenom immunoprofiling by antivenomics revealed differential antigenicity of venoms from *N. kaouthia* and *N. oxiana*, and poor reactivity towards 3FTxs and PLA_2_s. Moreover, there was considerable variation between antivenoms from different manufacturers. These data indicate that improvements to antivenom manufacturing in India are highly desirable.

## 1. Introduction

India leads the world in the annual number of snakebite deaths, with the latest available figures indicating around 56,000 deaths per year [1]. This approximates one death due to snakebite for every two due to AIDS, and mainly afflicts the most impoverished inhabitants of rural areas [2]. In addition, a much larger number of non-fatal bites lead to long-term disability and consequent economic hardship due to treatment costs and the long-term effects on the health and ability of survivors to work [3]. The size and biogeographical complexity of a country such as India, and the fact that herpetology has only recently become a popular field of study, means that the taxonomy and distribution of many species are still unclear. The presently available polyvalent antivenom is manufactured using the venom of only four species, the so-called “Big Four”: Russell’s viper (*Daboia russelii*), spectacled cobra (*Naja naja*), saw-scaled viper (*Echis carinatus*), and common Indian krait (*Bungarus caeruleus*), largely obtained from a single source, the Irula Snake Catchers’ Co-operative Society (henceforth referred to as the Irula Co-op), located south of Chennai in Chengalpattu District, Tamil Nadu. The lack of efficacy of the present polyvalent antivenom, caused by venom variation within and between species, has been well documented [4,5,6] and is a far from a trivial issue.

Among the venomous species prevalent in India, cobras are widespread and are responsible for significant morbidity and mortality [7,8,9]. All Asian cobras were once included in *N. naja*; however, the taxonomy of the Asian cobras was resolved in the early 1990s and *N. naja* was found to be restricted to India, Sri Lanka, Pakistan, Nepal, and Bangladesh [10,11,12]. There are three cobra species found in mainland India; in addition, the Andaman Islands have an endemic species *Naja sagittifera*, which is not considered here. The spectacled cobra *Naja naja* is the most commonly encountered venomous species across most of India, and the most widely distributed, found in all parts of India except the northeastern states. Despite being one of the Big Four species, there are still concerns about the effectiveness of antivenom in parts of India distant from the primary source of venom used in the manufacture of antivenom. Although typical neurotoxic symptoms of cobra bite are displayed, the extent of local necrosis varies, and in some cases, coagulopathy has been reported [13,14]. However, there is considerable evidence of variation in venom components in *N. naja* from different regions of India from proteomic, biochemical, pharmacological, and pathophysiological studies [15,16,17,18,19,20].

The monocled cobra, *Naja kaouthia*, co-occurs with *N. naja* in the northern Gangetic plain at least as far as western Uttar Pradesh, with an isolated record from Sonipat, Haryana [12]. The species is common in West Bengal and replaces *N. naja* altogether in the “Seven Sister” states (Arunachal Pradesh, most of Assam, Meghalaya, Manipur, Mizoram, Nagaland, and Tripura) of northeastern India, which are connected to the remainder of India by a narrow corridor separating Bangladesh from Bhutan and Nepal. Faunistically, these northeastern states are very distinct from the rest of India, sharing more species with the Indo-Malayan biogeographical region. This has serious consequences when it comes to snakebites, as none of the Big Four species included in currently available Indian antivenoms are found here (barring a small area of western Assam). Faiz et al. (2017) reported severe neurotoxicity as well as local blistering and necrosis resulting from bites by *N. kaouthia* in Chittagong Division, Bangladesh, similar to sequelae reported from bites by this species in Thailand [21,22]. Geographical variation in venom composition and activity has been reported from Southeast Asian populations of *N. kaouthia* from Malaysia, Thailand, and Vietnam [23,24,25]. However, although these are currently considered conspecific with the Indian populations, genetic evidence [26] and unpublished data suggest species-level differences between them. Deka et al. [27] recently reported on venom variation of this species from Assam and Bangladesh, whereas Das et al. [28] documented an LD_50_ of 0.148 mg/kg for *N. kaouthia* venom from Assam and showed that Indian polyvalent antivenom neutralized some tested biochemical and biological activities under in vitro conditions, albeit at high venom to antivenom ratios (1:100). Senji Laxmi et al. (2019) added data from Arunachal Pradesh and West Bengal, indicating that West Bengal venom displayed higher toxicity (0.18–0.28, mean 0.24 mg/kg) whereas Arunachal Pradesh venom displayed lower toxicity (1.14–1.33, mean 1.23 mg/kg) than venom from the previously described localities [29].

The Central Asian or Caspian cobra, *Naja oxiana*, was known from the literature to occur in the mountainous northwest of India, although at least some records resulted from misidentification of the black morph of *Naja naja*, which lacks the typical spectacle hood mark as an adult [11,12]. The first photographic and genetic confirmation of the occurrence of this species in India, from Chamba District in Himachal Pradesh, was provided recently [30]. Although the number of bites caused by this species is not known, they are unlikely to be high as the human population density in their Indian range is low. Nevertheless, the authors are aware of a case from Chamba District which required the victim to be given ventilatory support. A clinical report of the bite of this species from Afghanistan reports pain, erythema, edema, and coagulopathy that resolved on the administration of Favirept Polyvalent Snake Antivenin [31], and based on experiments on rabbits, Angaji et al. concluded that the venom of Iranian *N. oxiana* had acute effects on cardiac tissue during the first few hours following snake bite [32]. Latifi et al. reported the yield and LD_50_ of venom of the Caspian cobra from northern Iran to be variable between males and females, with males yielding more venom but with lower toxicity [33].

A large number of proteomic studies have been carried out by different laboratories using different methods, hence difficulty arises when the comparison between venom proteomes across *Naja* needs to be made [15,16]. We have addressed venom variation in Indian cobras at three levels: within-locality among individuals, within-species among localities, and within-locality among species. The former is rarely addressed since venom is often pooled from a number of individuals when milking, despite inter-individual variation being well documented in a number of species [34]. In order to rationalize snakebite treatment in the biogeographically distinct Greater and Lesser Himalayan regions, we further investigated the efficacy (cross-reactivity) of the available Indian antivenoms against species not currently included in their production.

## 2. Results

### 2.1. Sampling

Two samples of *N. naja* and one of *N. oxiana* from Himachal Pradesh, three *N. naja* and one *N. kaouthia* from West Bengal, one *N. kaouthia* from Mizoram, Assam and Arunachal Pradesh, and two *N. naja* from Maharashtra were analyzed (Figure 1). Adult males were used for comparison to guard against possible sexual dimorphism and ontogenetic variation [35,36]; however, the sex of the individual from Assam was not known.

### 2.2. Comparative Analysis of HPLC Profiles Reveals Variation

Within-locality among individuals: The HPLC protein profiles of adult *N. naja* individuals from the same localities of West Bengal and Maharashtra were found to be identical in terms of protein peaks eluted, but remarkable differences in the relative absorbance of the peaks were observed (Figure 2A). However, individual *N. naja* venoms sourced from Chamba District of Himachal Pradesh presented similar profiles under identical chromatographic conditions.

Within-species among localities: The chromatograms of *N. naja* venom samples from three different locations (Himachal Pradesh, Maharashtra, West Bengal) exhibited considerable differences in peak intensities and the number of protein peaks (Figure 2A). For example, the abundance of protein peaks eluting in the initial 40 min (solvent gradient 20–23%) was greatly enhanced in the *N. naja* sample from Himachal Pradesh. On the other hand, the HPLC pattern of the sample from West Bengal was markedly different from the other three venoms, with a greater number of protein peaks emerging between 40 and 60 min at a solvent gradient of 27–30%. Moreover, substantial differences in elution profiles were observed when compared to venom samples from the Irula Co-op from Tamil Nadu.

The venom elution profiles were much more similar among *N. kaouthia* samples from four different locations: Arunachal Pradesh, Mizoram, West Bengal, and Assam (Figure 2B), with the HPLC profiles of *N. kaouthia* venoms from Arunachal Pradesh and Mizoram appearing relatively uniform. The HPLC profiles of *N. kaouthia* venom samples revealed that the majority of proteins eluted before 100 min (solvent gradient 20–43%). The venom samples demonstrated larger areas of HPLC peaks eluting between a retention time of 60 and 100 min. A distinct HPLC peak eluting at a solvent gradient of 27–30% was specifically observed in the case of the Assam sample and was absent from other samples. However, it should be noted that the sex of the Assam specimen was not known, so these differences may be due to sexual dimorphism.

Among different *Naja* species: The venoms of all *Naja* species were analyzed using the same HPLC protocol previously standardized by Deka et al. [37]. Its use in proteomics analysis of venom of *N. kaouthia* from Assam revealed that 3FTxs and PLA_2_s elute within the region indicated by a dotted box in Figure 2. The chromatographic profiles of *N. naja* samples varied from other *Naja* species in terms of the number of protein peaks and their abundances within this region. The elution pattern of the *N. oxiana* venom sample appeared much more similar to *N. kaouthia* samples (Figure 2B,C). Interestingly, the unique peak observed in the case of the *N. kaouthia* sample from Assam (starred in Figure 2B) was also present in the *N. oxiana* sample.

### 2.3. Venom Proteomes of Different Naja Venoms

*Naja naja* venoms: The relative abundances of different protein families identified in the venoms of *Naja naja* from Maharashtra, West Bengal, Himachal Pradesh, and Tamil Nadu are shown in Figure 3. Overall estimation of relative abundances (in the percentage of total proteins) indicated three-finger toxins (3FTx: 58–91%) and phospholipase A_2_ enzymes (PLA_2_: 4–26%) as the major protein families present. Although cardiotoxins (CTx) belong to the 3FTx family, they are represented separately in Figure 4. Low abundance proteins in *N. naja* venoms, including cysteine-rich secretory proteins (CRISP), l-amino acid oxidases (LAAO), nerve-growth factors (NGF), nucleotidases, phosphodiesterases (PDE), acetylcholinesterase, serine proteases, and tissue plasminogens were present in the venoms at an abundance less than 1%, and are labeled as ‘Others’ in Figure 4. Detailed analysis of protein profiles of seven *N. naja* venoms revealed that the 3FTxs and PLA_2_s are made up of similar protein isoforms. In particular, each venom analyzed displayed the presence of 3FTxs similar to *N. naja* long neurotoxin P25668 (Appendix A). Proteomic comparisons among individuals within the same locality were performed for *Naja naja* venoms from Maharashtra and West Bengal. Considerable variations in the relative abundances of toxin families were seen among the individuals from both locations. In the case of individuals from Maharashtra, cardiotoxins were detected in trace amounts (<0.1%) in Sample B, in contrast to Sample A (6%) (Figure 3, sample A. In the case of West Bengal individuals, the PLA_2_ toxins showed variation in abundance. The Venn diagram (Figure 3) showed that individuals from Maharashtra have around 50% of peptides in common whereas in West Bengal, 20% of shared peptides were detected.

*Naja kaouthia* venoms: Three out of four venoms studied here were subjected to protein identification by mass spectrometry. The proteomics analysis revealed that the venom proteome of *Naja kaouthia* is mainly dominated by 3FTx (largely comprised of cardiotoxins), followed by PLA_2_, together accounting for around 96% of total venom proteins (Figure 4I–K). The percentage abundance of cardiotoxins was higher than in *N. naja* venoms. Nerve growth factor, CRISP, natriuretic peptide, nucleotidase, phosphodiesterase, and LAAO were detected in minor amounts (labeled as ‘Others’). Based on our analysis, 35 proteins belonging to 9 different protein families were detected in the venom sample from Mizoram. Mass spectrometry of samples from Arunachal Pradesh and West Bengal identified 34 (12 families) and 42 proteins (14 families), respectively. Acetylcholinesterase, tissue plasminogen activator, venom factor, and vespryn were detected in trace amounts only in the West Bengal venom samples, whereas we detected serine protease and Kunitz-type inhibitors only in the Arunachal Pradesh sample.

*Naja oxiana* venom: The venom toxin composition for *Naja oxiana* venom was unique among the *Naja* species studied. The dominant toxin families included 3FTx (85.3%) and SVMP (9.1%). Cardiotoxins were present in trace amounts (0.1%). Other low-abundance toxin families such as 5′-nucleotidases, CRISP, Cobra venom factor, L-amino acid oxidase, and phosphodiesterases were also detected.

Comparison of proteome profiles among different *Naja* species: A close comparison of protein profiles suggested that *Naja* species share a number of highly similar 3FTxs. For instance, the venoms of both *N. kaouthia* and *N. oxiana* contain peptide fragments that exhibit amino acid sequence similarity with weak neurotoxin P25679. Peptide fragments bearing sequence similarity with long neurotoxin P25668 were detected in 9 venoms out of the 11 samples analyzed. The venom proteomes of two *Naja* species (*N. naja* and *N. oxiana*) from Himachal Pradesh revealed comparable relative abundances of 3FTxs (86–88%). As evident from proteome profiles, *N. naja* and *N. kaouthia* venoms from West Bengal exhibited qualitative differences in venom composition, which is further supported by a previous report [38]. Higher numbers of unique peptides were observed for *N. kaouthia* and *N. naja* from West Bengal (Figure 5). We observed 17% shared peptides across the genus (Figure 6).

### 2.4. Cross-Reactivity of Indian Polyvalent Antivenoms

The electrophoretic patterns of *Naja* samples collected from different locations were investigated (Figure 7A). Separation of venom proteins under reducing conditions revealed the presence of protein bands ranging between 10 and >40 kDa, with significant variation in protein band intensities. The SDS-PAGE profile in all venom samples revealed an abundance of protein bands in the 10–15 kDa molecular mass region. A high-intensity band at ~10 kDa was observed in all samples, whereas the intensity of the protein band at 15 kDa was highly variable between the samples. In addition, a few protein bands in the higher molecular mass region (>40 kDa) were observed.

Assessment of cross-reactivity of two Indian polyvalent antivenoms by Western blot revealed the ability of both to recognize higher molecular weight venom proteins of *N. oxiana* (Figure 4B,C), with lesser recognition of lower molecular weight proteins. VINS antivenom exhibited poor reactivity towards the low molecular mass protein bands. In contrast, Premium Serums displayed greater cross-reactivity towards the low molecular mass proteins of *Naja* venoms. Strikingly, VINS antibodies displayed extremely low reactivity towards high molecular weight proteins (≥25 kDa) of *N. naja* samples from locations other than that of the Irula Co-op in Tamil Nadu.

Phospholipase activity was significantly different among samples, specifically being highest in the *N. naja* venom from the Irula Co-op in South India, and lowest in *N. oxiana* from Himachal Pradesh (Figure 8A). *Naja kaouthia* venom from Arunachal Pradesh also showed reduced activity compared to the venom from the same species from other locations. In all cases, the phospholipase activity of *N. kaouthia* and *N. oxiana* venom was reduced to negligible levels by incubation with tested polyvalent antivenoms (Figure 8B). In contrast, PLA_2_ activities of *N. naja* venoms were not effectively neutralized by either antivenom despite being the species included in the immunizing mixture, particularly in the case of *N. naja* from Himachal Pradesh.

### 2.5. Immunoaffinity Chromatographic Profiling of Indian Polyvalent Antivenoms

An antivenomics approach was employed to investigate the immunocapturing ability of two Indian polyvalent antivenoms (VINS and Premium Serums) towards heterologous venom proteins of *N. kaouthia* and *N. oxiana*. Figure 9 depicts the outcome of antivenomic analyses of VINS PAV and Premium Serums PAV towards the venoms of *N. oxiana* venom (Himachal Pradesh) and *N. kaouthia* venoms (Arunachal Pradesh, Mizoram, West Bengal). When assessed with 75 µg of *N. kaouthia* and *N. oxiana* venoms, both the antivenoms retained the majority of venom proteins, although to a varying extent. The analysis of venom proteins retained and non-retained in the antivenom affinity columns clearly indicates that quantitative differences exist in immunorecognition capacity between VINS and Premium Serums antivenoms. For example, VINS PAV-immobilized affinity columns retained around 80% of the total input of *N. oxiana* venom whereas 72% of venom proteins were retained in the Premium Serums PAV-immobilized affinity column. When antivenom affinity columns were loaded with *N. kaouthia* venoms, higher binding affinity towards the proteins of the sample from Arunachal Pradesh was observed (77% for VINS PAV; 80% for Premium Serums PAV). The percentage of proteins immunoretained in the Premium Serums PAV affinity column was lowest (~46% of total input) for *N. kaouthia* venom sourced from Mizoram.

Figure 9C,F,I,L illustrate the percentage of each peak retained and non-retained by immunoaffinity columns with VINS and Premium Serums PAV. Based on the antivenomic profiles, both antivenoms possess good immunorecognition of toxins in the venom of *N. oxiana* (Figure 9A). However, one major peak was not retained while two were partially retained. In the case of the *N. kaouthia* samples, both the antivenoms retained the protein peaks only to a partial extent (Figure 9D). These antivenoms were less efficient in immunocapturing the proteins contained in peak F1.

### 2.6. Identification of Non-Immunoretained Proteins

Only those non-retained peaks with high protein concentrations were subjected to protein identification by mass spectrometry. Appendix A lists proteins identified in non-retained fractions of *N. oxiana* and *N. kaouthia* venoms. Samples from Arunachal Pradesh and Mizoram had the highest number of non-retained toxins identified. For *N. kaouthia* venom from Arunachal Pradesh, 20 3FTxs (including 4 CTxs), as well as one protein each corresponding to PLA_2_, LAAO, and NGF were identified in the non-immunodepleted retained fraction (Figure 6; Appendix A). In the case of the Mizoram sample, 18 3FTxs (including 3 CTxs) and 9 PLA_2_s were identified, whereas 4 3FTxs (including 1 CTx) and 2 PLA_2_s were identified in *N. kaouthia* from West Bengal. For the *N. oxiana* sample, 10 3FTxs (including 1 CTx) were identified.

## 3. Discussion

Venom variability is a ubiquitous phenomenon and occurs at different taxonomic levels, including intraspecific [40]. This high degree of venom variability is the result of gene duplication and functional divergence that has led to the rapid evolution of venom proteins [41,42]. The implications of venom variation on pathological effects and management of snakebite are of serious concern and need urgent attention [18,24,43,44,45,46,47]. Thus, understanding the venom proteome is instrumental in providing useful information on unique toxins and proteins in snake venoms, and interpreting the clinical symptoms during envenomation. With advances in proteomic technologies, studies unraveling the complexity of the venom proteome have become more feasible. Cobras of the genus *Naja* are listed in the World Health Organization’s (WHO) category I of medically important snakes, and several proteomics studies have suggested that venom composition differs by geographical origin [48,49]. Such intraspecific venom variability is accompanied by varied envenoming effects and differential neutralization efficacy of existing antivenoms. In the present study, compositional variation has been evaluated for venoms from Indian *N. naja*, *N. kaouthia*, and *N. oxiana* at several levels: among *Naja* species, within *Naja* species from different geographical locations, and individual variation among specimens of the same species.

Several proteomics studies on *N. naja* venoms have been carried out on pooled venoms [16,39,50,51] despite the heterogeneity of venoms at an individual level being well documented in the literature and occurring in many snake species. In this study, reverse-phase HPLC protein profiles of individual *N. naja* specimens from the same localities were examined. Profiles exhibited similar patterns among the specimens; however, protein concentration varied between individuals from the same locations. Modahl and colleagues investigated the compositional profiles of captive *N. naja* populations and found similar venom composition but with slight variability in isoform concentration [52]. Huang et al. [49] reported wide differences in proteomic and glycomic profiles in adult individuals of *N. atra*. The researchers concluded that proteomics analysis to address inter-individual variations could be highly relevant for better antivenom production.

Accordingly, proteomics studies were carried out to explore the compositional disparity in individual *N. naja* specimens. In line with SDS-PAGE and RP-HPLC profiles, proteomics analysis of individual specimens exhibited noticeable differences in the expression of protein components. Unexpectedly, we observed more dramatic differences in relative abundances of protein families between *N. naja* individuals from West Bengal than among the specimens collected from other geographical locations. Inconsistency in the relative percentage of proteins was also observed when two different laboratories performed venom proteomics of West Bengal samples [16,29,38]. Several factors including sexual dimorphism and ontogeny might contribute to such heterogeneity of venoms but have been controlled for in this study. When the proteome profile of *N. naja* venom procured from the Irula Co-op (Tamil Nadu), which is largely used for the production of polyvalent antivenoms, was compared with heterologous *N. naja* venoms, stark differences were observed in the toxin abundance. Such differences at the intra-specific level provide a reasonable explanation for the differences in lethal potencies and variable antivenom neutralization efficacies observed [16,29,38].

The most abundant and lethal components in cobra venoms are three-finger toxins. These non-enzymatic toxins are known to induce a range of pharmacological effects including neurotoxicity, cytotoxicity, and hemostasis dysfunction. In the present study, the abundance of 3FTxs varied significantly within *N. naja* specimens from different geographical locations. Previous analysis of 3FTx composition of *N. naja* venoms sampled from Western India (Maharashtra) revealed considerable differences within a small geographical range [29,51]. In addition, a comparison of 3FTx content in West Bengal samples in this study with findings reported by others [50] suggests remarkable differences in the relative abundances. Moreover, variations in 3FTx profiles are known to be reflected in the toxicity of *Naja* venoms [23,53]. Accordingly, it is very likely that the marked compositional differences among populations within a small geographical range contribute to the differential reactivity observed with existing antivenoms. The observed compositional variation may also be influenced by differences in the timing of the last meal among individuals [54], and the variable expression time of each toxin during replenishment, but this requires further study.

Several high-throughput proteomics strategies have been used to characterize *N. kaouthia* venoms from different geographical regions of Southeast Asia [23,55]. Using multidimensional proteomics methods, Deka et al. [27] interpreted qualitative differences in venom composition in *N. kaouthia* samples from two geographically adjacent regions, Northeast India and Bangladesh. A recent study reported venom proteome analysis of two *N. kaouthia* populations from India, but a more comprehensive analysis of venoms from different populations is vital to elucidate biogeographical variation [29]. Therefore, we investigated the protein profiling of *N. kaouthia* venoms sourced from the eastern (West Bengal) and northeastern regions of India (Arunachal Pradesh, Mizoram). Interestingly, the proteomics data of analyzed *N. kaouthia* venoms revealed comparable relative abundance values for 3FTxs as well as PLA_2_s between *N. kaouthia* venoms. We observed the presence of cardiotoxins in high abundance. In contrast to our findings, Senji Laxme et al. [29] reported a greater quantity of 3FTxs in *N. kaouthia* venom from West Bengal than from Arunachal Pradesh. Notably, significant differences in venom composition and biochemical activities were recently reported in *N. kaouthia* venoms sampled at a relatively small spatial scale in Eastern India [34]. Hence, a more comprehensive study on inter-individual and intra-population venom variation in *N. kaouthia* is warranted to provide insights into its impact on the effectiveness of existing antivenoms.

A comparison between the proteome profiles of *N. kaouthia* and *N. naja* from the same locality in West Bengal revealed remarkable quantitative differences in venom proteins. In both venoms, unique proteins accounted for ~50% of total proteins, which is in agreement with a previous study [37]. To the best of our knowledge, this is the first study to report proteomics of *N. oxiana* venom of Indian origin and to compare their venom proteome with congeneric *Naja* species. We found *N. oxiana* venom to be composed of diverse pharmacologically active protein components such as 3FTxs, SVMPs, PLA_2_s, and several other minor components (CRISPs, venom factor, LAAOs, phosphodiesterases, etc.). A recent study investigated the proteome of Pakistani *N. oxiana* using a shot-gun proteomic approach [56] and reported an equal abundance of 3FTxs and SVMPs which together accounted for approximately 32% of the total venom proteome. However, our study revealed a much higher percentage of 3FTxs (~89%) than SVMPs (~7%) and a much lower percentage of PLA_2_s (~2% of total proteins). The notable differences observed in the relative abundances of venom proteins of *N. oxiana* from India and Pakistan may be due to individual variation, but may also account for differences in pathophysiological effects in envenomated victims. Comparative analysis of venoms of *N. naja* and *N. oxiana* from Himachal Pradesh displayed substantial variation in venom proteomes. A similar observation was also made in a study investigating venom proteome profiles of Pakistani *N. naja* and *N. oxiana* [57].

Even though extensive inter- and intra-specific venom variation exist, clinicians are obliged to use polyvalent antivenoms raised against venoms of the Big Four from populations inhabiting a restricted area in Tamil Nadu, collected by the Irula Co-operative Society. Large amounts of antivenom are frequently administered to snakebite victims, raising the risk of adverse reactions, increasing the cost of treating snakebites, and decreasing confidence among doctors and patients in the use of antivenom. Several reports have suggested differences in antivenom efficacy when used against snakebites in regions far from the source of immunizing venoms [9,18,34,58,59,60]. Moreover, clinical studies investigating the efficacy of Indian antivenoms in the treatment of envenoming caused by venomous species other than the Big Four have reported the ineffectiveness of heterologous polyvalent antivenoms and their potential risk in causing adverse reactions [21,61,62,63]. The evaluation of the cross-neutralization potential of antivenoms against venoms not included in the immunization mixture may help in expanding the range of existing antivenoms for clinical use. In this work, we studied the cross-neutralization potential of Indian polyvalent antivenoms against the heterologous venoms not used in immunization mixtures. Analysis of immunoblots in this study revealed that antibodies in both the antivenoms cross-reacted with high molecular mass *N. kaouthia* and *N. oxiana* venom proteins (≥25 kDa) more effectively. These are likely to belong to cysteine-rich secretory protein (CRISP) and metalloproteinase families (SVMPs). The Western blot results of Premium Serums PAV towards *N. naja* venom samples are in line with previous studies [16,38,51]. However, VINS polyvalent antivenom notably displayed very weak cross-reactivity towards small proteins (Mol wt. ≤ 15 kDa) from *N. kaouthia* venoms. The low immunogenicity of these smaller venom toxins likely prevents the production of a sufficient titer of toxin-specific antibodies [64,65]. Recombinant antibody formats have been proposed as an alternative to conventional antivenoms for effective neutralization of the weakly immunogenic venom components but have not yet been clinically deployed [66,67].

In vivo venom neutralization assays with antivenom remain the standard test for determining antivenom efficacy [68]. However, prior investigations using in vitro assays to estimate the neutralization efficacy of antivenoms are recommended to reduce the use of mice [55] and the data presented here can be used in further in vivo studies. Antivenomics profiles of Indian polyvalent antivenoms revealed differential patterns of immunorecognition towards different protein peaks when loaded with a fixed venom dose. The most remarkable observation made was that majority of unbound proteins belong to 3FTx and PLA_2_ families, which are major toxic components of *N. kaouthia* and *N. oxiana* venoms. Similar observations have also been made in previous studies, implicating the presence of both non-antigenic and unique proteins in the venoms [27,37,38]. The findings in our study explicitly address the correlation between protein molecular mass and immunodepletion, with the weakest depletions for low molecular mass proteins. Although both antivenoms tested performed well in neutralizing in vitro PLA_2_ activities of venoms from *N. kaouthia* and *N. oxiana*, they were largely ineffective towards PLA_2_s of *N. naja*. The results from the neutralization of biochemical activities of *N. naja* and *N. kaouthia* venoms corroborate well with the previous outcomes based on antivenom neutralization assays [39,51]. Although the presence of similar PLA_2_ isoforms was reported among the studied *N. naja* proteomes, significant variation in the percentage composition suggests differences in the expression of these shared PLA_2_ genes. Such quantitative differences could not only account for variable envenoming manifestations but may also be responsible for the poor performance of polyvalent antivenoms in the neutralization of the PLA_2_ activity of *N. naja* venoms. The proteomics data showed higher PLA_2_ abundance in *N. kaouthia* venoms, but fewer isoforms, compared to *N. naja*. Hence, the better neutralization of *N. kaouthia* could be because of lower isoform diversity in their venoms but also because the PLA_2_ isoforms from *N. kaouthia* venoms are also found in the *N. naja* venom from the Irula Co-op.

An immunocapturing capability of 20–25% of total proteins is often suggestive of good outcomes in in vivo neutralization assays [69]. The antivenomics data in the present study corroborates well with the results of previous invivo neutralization assays, suggesting the cross-reactivity of Indian polyvalent antivenoms towards heterologous venoms [27]. As earlier reported, Premium Serums antivenom (Batch No. 212013) displayed neutralization potency towards *N. kaouthia* venom from West Bengal (ED50 value, 0.156 mg/mL) while completely failing to neutralize the lethal effects of *N. kaouthia* venom from Arunachal Pradesh (Senji Laxme et al., 2019) [29]. Moreover, VINS polyvalent antivenom was found to be effective in neutralizing the lethal effects of *N. kaouthia* venom from Northeast India (ED50 value, 76.38 venom mg/antivenom g) [27]. These studies perhaps indirectly support our observation that polyvalent antivenoms partially immunorecognized venom proteins of the tested samples. Overall, our findings signify the existence of common antigenic determinants among the venoms of *N. naja*, *N. kaouthia*, and *N. oxiana* that potentially explain why the Indian polyvalent antivenom displayed a good immunocapturing ability of venom proteins. However, it is worth noting that the overloading of venom proteins decreases the binding affinities to the immunoaffinity column and results in the appearance of proteins in both retained and non-retained fractions [27,38,70]. The fact that 200 µg venom can exceed the immunocapturing ability of 5 mg of antivenom is concerning, even if recognition of proteins is good at lower doses. Cobras are capable of injecting large volumes of venom in a single bite; hence, it is likely that many vials of antivenom would be needed for the neutralization of toxins injected in an average bite, increasing the cost of treatment and the chances of adverse reactions such as anaphylaxis and serum sickness.

Thus, our proteomics data, in vitro neutralization, and antivenomics analyses highlight the need to redesign the immunization mixture for the manufacture of antivenom for use in India and emphasize the inclusion of adequate pools of venom from different regions and species, or development of regional antivenoms, thereby generating more effective treatment of bites. Although the present study highlights good reactivity of polyvalent antivenom towards heterologous venoms of three *Naja* species, despite observed proteomic differences, identification of key medically relevant toxins in the whole venom and their inclusion in immunization mixtures would serve as a valuable solution for the improvement of snakebite treatment.

From an evolutionary point of view, the huge variation in PLA_2_ content and activity, especially at the individual and population level in *Naja naja*, is noteworthy given the important role of PLA_2_ in causing an algesic effect and its relation to spitting behavior [71]. The proportion of PLA_2_ present in both *N. naja* and *N. kaouthia* in our analyses are considerably different from those in Kazandjian et al. [71] and sound a cautionary note for evolutionary inferences based on a limited sampling of variation at the individual and population level.

## 4. Materials and Methods

### 4.1. Venoms and Antivenoms

In Himachal Pradesh and Mizoram, visual encounter surveys [72] were undertaken by day and night, with effort focused on sampling areas that appeared to provide species-specific optimum habitat. These were supplemented by active searches, involving turning rocks and logs, peeling bark, digging through leaf litter, and excavating burrows and termite mounds. Elsewhere, snakes were primarily obtained from rescuers immediately after removal from a human–snake conflict situation. Snakes were retained in securely tied cotton bags until processed, which took place as soon as practical after capture. Snakes were restrained in clear plastic tubes to reduce stress and for the safety of handlers [73]. The head was allowed to protrude from the end of the tube, where it was offered a plastic-film-covered Nalgene beaker to bite. No massaging of venom glands was carried out. Snakes were released in a suitable location, as close to the site of capture as practical, after sampling. Venom was then transferred to one or more 1.5 mL polypropylene tubes and dried for storage at −20 °C until further analysis. *Naja naja* venom purchased from the Irula Co-op was used as a standard comparison, as this is the primary source of venom used to manufacture antivenom in India. Apart from the Irula venom, samples were not pooled and were analyzed individually. Details of venom collection are provided in Table 1. The protein concentrations of venoms were estimated according to the colorimetric method developed by Lowry [74]. Bovine serum albumin (Sigma-Aldrich, St Louis, MO, USA) was used as the protein standard for the quantification of protein in venom.

Antivenoms used in this study were manufactured by VINS Bioproducts Ltd., Hyderabad, India (Batch No 01AS15007; Expiry date: January/2019) and Premium Serums & Vaccines Pvt Ltd., Narayangaon, Maharashtra, India (Batch No. 212013; Expiry date: August/2020). One vial of each antivenom was reconstituted in 3 mL MilliQ and the amount of F(ab’)_2_ fragments in the polyvalent antivenoms were quantified using a Nanodrop 2000 spectrophotometer (Thermo Fisher Scientific, Waltham, MA, USA) with the protein A_280_ method using the IgG mass extinction coefficient. Measurements of protein concentration in antivenoms were performed three times. Table 2 displays the characteristics of Indian polyvalent antivenoms used in the present study.

### 4.2. Venom Protein Profiling Using Reverse-Phase HPLC

Crude venoms (2 mg) were reconstituted in ultrapure water, centrifuged to remove debris, and separated by reverse phase HPLC using Symmetry C18 column (250 × 4.6 mm, 5 μm particle size, 300 Å pore size) (Waters, Milford, MA, USA) and Thermo Scientific UltiMate 3000 high-performance gradient system equipped with a Diode-Array Detector (Thermo Fisher Scientific, Waltham, MA, USA). Elution was carried out at a flow rate of 1 mL/min using Solution A containing 0.1% (*v/v*) trifluoroacetic acid (TFA) and Solution B containing 0.1% TFA in 80% acetonitrile. The gradient used for the separation of venom proteins was 20–55% Solution B for 100 min and 55–80% for 20 min. Protein detection was performed spectrophotometrically at 215 nm and fractions were collected for further analysis. As a comparator, the venom sample of *N. naja* (Irula Co-op, Vadanemmeli, Tamil Nadu, India) was also subjected to reverse-phase HPLC using the same chromatographic conditions.

### 4.3. Whole Venom in-Solution Tryptic Digestion and Protein Identification by Mass Spectrometry

A total of 10 µg of venom samples were prepared for analysis by breaking the proteins into peptides using in-solution trypsin digestion following protocols adapted from Kintin and Sherman [75] and Calvete [76]. All the reagents used in this methodology were procured from Sigma Aldrich, St Louis, Missouri, USA. Fractions were reconstituted in 10 µL 6 M Urea, 50 mM Tris-HCl, pH 8.0, reduced with 0.5 µL 200 mM DTT, and alkylated with 2 µL 200 mM Iodoacetamide dissolved in 50 mM Tris-HCl, pH 8.0. Samples were then diluted with 77.5 µL Tris-HCl, pH 8.0 to reduce the concentration of Urea to 0.6 M. Subsequently, 100 ng Trypsin (Roche Diagnostics) was added to each sample and incubated at 37 °C for 12 h. Formic acid was added to stop the reaction and adjust the sample pH to between 3 and 4. Samples were desalted using C-18 Ziptips (Thermo Fisher Scientific, Waltham, MA, USA) and digested peptides were eluted in 70% ACN, 0.1% TFA. The digested peptides were loaded on a nanospray capillary column (PepMapTM RSLC C18, Thermo Fisher Scientific, Waltham, MA, USA) and subjected to sequencing by MS/MS in Q-Exactive HF mass spectrometer (Thermo Fisher Scientific, Waltham, MA, USA). The spectra were analyzed using Proteome Discoverer (Version 2.2), using the workflow described in Appendix A, against the NCBI Taxonomy database from Elapidae (Taxonomy ID: 8602). MS/MS mass tolerance was set to 10 ppm. Carbamidomethyl cysteine was set as a fixed modification, and oxidation of methionine and deamidation of Arginine and Glutamine were set as variable modifications. Doubly and triply charged peptides were selected and matched proteins were filtered for at least 2 unique peptides. Individual venom protein family abundance was calculated based on its mean spectral intensity (MSI) relative to the total spectral intensity of all proteins detected [77].

### 4.4. Cross-Neutralization of Venom PLA_2_ by Polyvalent Antivenoms

Phospholipase A_2_ activities of *N. kaouthia*, *N. naja*, and *N. oxiana* venoms were determined with an sPLA_2_ assay kit (Cayman Chemical Company, Ann Arbor, MI, USA). PLA_2_ activity was expressed as micromoles of phosphatidylcholine hydrolyzed per min per μg per enzyme. For neutralization assay, a constant venom dose from each sample was pre-incubated with antivenoms for 1 h at 37 °C such that the venom to antivenom ratio in the reaction mixture was 1:100 (protein/protein). Following incubation, the PLA_2_ activity of venom samples was assessed using the method previously described [38]. Activities of crude venoms in absence of antivenoms were taken as negative controls. For the non-enzymatic control, all the reagents except for venom were added to the well. The percentage neutralization was calculated considering the activity of the venom samples in absence of antivenom as 100%. Experiments were carried out in triplicate and the results were presented as mean ± standard deviation.

### 4.5. Immunochemical Analysis

Briefly, 20 μg of reduced venom samples were loaded onto the gel and electrophoresis was carried out. The gels were either stained with Coomassie brilliant blue R250 (Merck, Darmstadt, Germany) or were subjected to Western blot as described previously [27]. Dilution ratios for tested antivenoms were kept constant (1:5000; *v*/*v*) to ensure a valid comparison between antivenom and venom reactivity. The membrane was washed thoroughly with TBST and probed with anti-Horse IgG alkaline phosphatase produced in rabbits (Sigma). For color development, BCIP/NBT liquid substrate (Sigma)was used. Images were documented with Chemidoc XRS imaging system (Biorad, Hercules, CA, USA).

### 4.6. Immunological Profiling of Polyvalent Antivenoms Using an Antivenomics Approach

The immunoaffinity column was prepared following the protocol as previously described [68,78]. For preparing affinity chromatography columns, 300 µL of CNBr activated Sepharose™ 4B matrix (GE Healthcare, Buckinghamshire, UK) was packed in a Pierce centrifuge column (Thermo Fisher Scientific, Waltham, MA, USA) [37]. The Sepharose beads were activated with 10 matrix volumes of ice-cold 1 mM HCl followed by washing with 2 matrix volumes of coupling buffer (200 mM NaHCO_3_ containing 500 mM NaCl, pH 8.5). Antivenoms were dialyzed against water using SnakeSkin Dialysis Tubing, 3.5K MWCO (Spectrum Laboratories, Rancho Dominguez, CA, USA). Antivenoms were then lyophilized and reconstituted in a coupling buffer. The dialyzed antivenoms were added to activated beads and the suspension was incubated overnight at 4 °C with gentle shaking. The amount of antivenom coupled to beads was quantified by measuring the concentration before and after incubation using a Nanodrop 2000 (Thermo Fisher Scientific, Waltham, MA, USA). Coupling yields were 5 mg for VINS polyvalent antivenom and 4.9 mg for Premium Serums. To block the remaining activated sites, the affinity columns were incubated with 0.1 M Tris-HCl, pH 8.5 for 4 h at room temperature. Following incubation, columns were washed alternatively with 900 µL of low pH buffer (0.1 M acetate buffer containing 0.5 M NaCl, pH 4.0) and 900 µL of high pH buffer (0.1 M Tris-HCl, pH 8.5). After repeating this procedure six times, the columns were equilibrated with five matrix volumes of binding buffer (20 mM phosphate buffer, 135 mM NaCl, pH 7.4; PBS). For the immunoaffinity assay, *N. kaouthia* and *N. oxiana* venom were incubated with the columns for 1 h at 25 °C with gentle shaking. At first, the optimal venom protein load for the immunoaffinity column was determined by incubating the affinity matrix with increasing amounts (75, 125, 200 µg) of *N. kaouthia* venom (Arunachal Pradesh). The abundance of bound proteins to the affinity column reached saturation at higher venom concentration (125, 200 µg) suggesting that non-specific binding of venom proteins to antibodies occurs at this concentration (Appendix A). Therefore, the antivenomics experiment was performed with 75 µg crude *Naja* venom and ~5 mg antivenoms, corresponding to a venom-to-antivenom ratio (m/m) of 1:67. As a specificity control, 300 µL matrix (without antivenom) was incubated with venom. Non-retained fractions were collected with 5 matrix volumes of PBS and retained fractions were obtained with 5 matrixvolumes of 0.1 M glycine-HCl, pH 2.0, and immediately neutralized with 150 µL1 M Tris-HCl, pH 9.0.

Non-retained and retained fractions were separately desalted and concentrated by using Pierce Protein concentrators (3 kDa MWCO), reconstituted in MilliQ, and the protein concentration quantified using a Nanodrop 2000 (Thermo Fisher Scientific, Waltham, MA, USA). The concentrated fractions were reconstituted in ultrapure water and the protein contents of the fractions were quantified. Non-retained and retained fractions were separately subjected to Thermo Scientific Acclaim C18 reversed-phase column (150 × 2.1 mm, 3 μm particle size, 300 Å pore size) using a Thermo Scientific UltiMate 3000 high-performance gradient system equipped with diode detector. The chromatographic peak areas of non-retained and retained fractions were determined using Dionex Chromeleon 6.8 software (Thermo Fisher Scientific, Waltham, MA, USA). The chromatographic areas were integrated manually to determine the total percentage of venom proteins unbound to the affinity column. Relative amounts of non-retained molecules (%NR_i_) were calculated as 100 − [(R_i_/(R_i_ + NR_i_)) × 100], where R_i_ corresponds to the chromatographic area of peak “i” in the chromatogram of fraction retained and eluted from affinity column [69].

### 4.7. Identification of Non-Retained Proteins by Tandem Mass Spectrometry

Toxins in the non-retained fractions obtained from immunoaffinity columns were subjected to proteomics analysis. Samples were prepared for analysis by breaking the proteins into peptides using in-solution trypsin digestion following the same protocol described in Section 4.3.

## Figures and Tables

**Figure 1 toxins-15-00258-f001:**
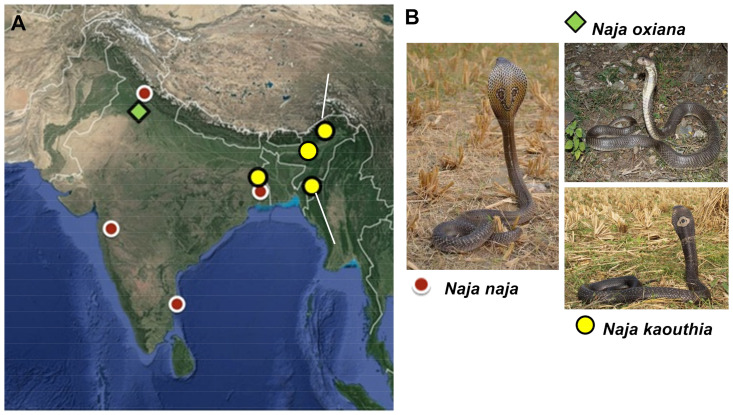
(**A**) Physical map of the Indian subcontinent showing sampling locations of *Naja* venoms investigated in this study. (**B**) Photographs of three different *Naja* species found in India (photographs by Vishal Santra).

**Figure 2 toxins-15-00258-f002:**
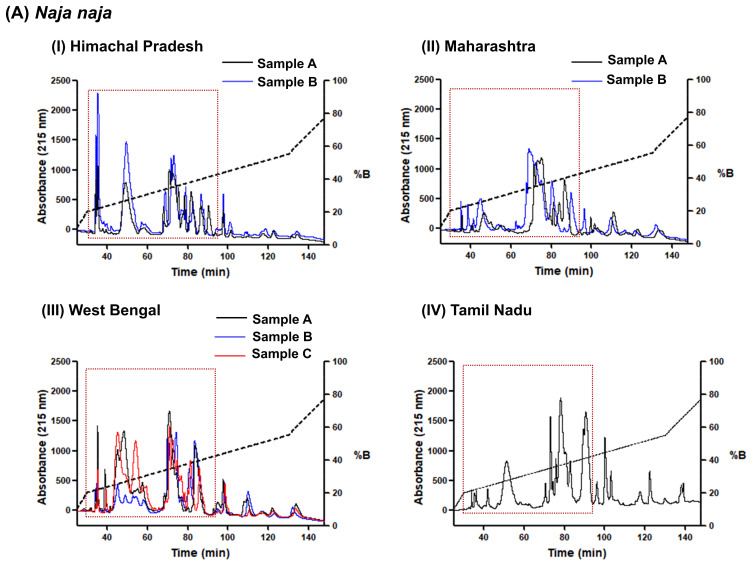
C18 reverse-phase HPLC profiles of venoms from three different *Naja* species in India. The dotted areas indicate the elution region of 3FTxs and PLA_2_s in *Naja* venoms ((**A**): *N. naja*, (**B**): *N. kaouthia*, (**C**): *N. oxiana*) in this standardized chromatographic gradient. The red star identifies a peak present only in the venom from Assam (see text for further discussion).

**Figure 3 toxins-15-00258-f003:**
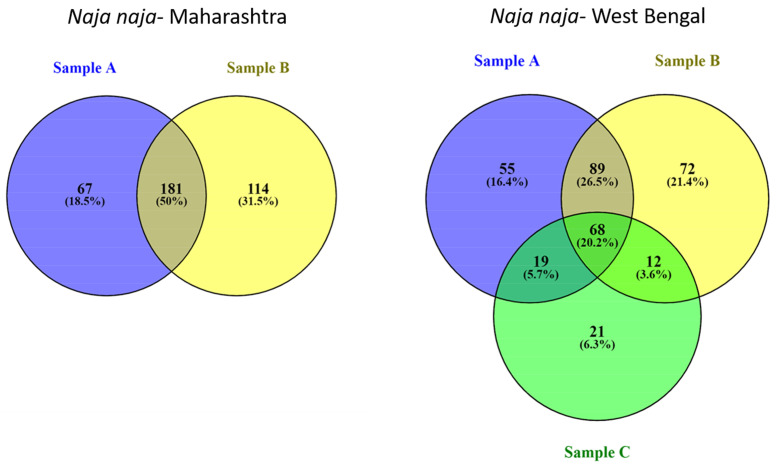
Venn diagrams representing venom variation among individuals from the same locality were constructed using Venny 2.1. Peptides were given unique IDs for their sequences and type of modifications from MS/MS spectra.

**Figure 4 toxins-15-00258-f004:**
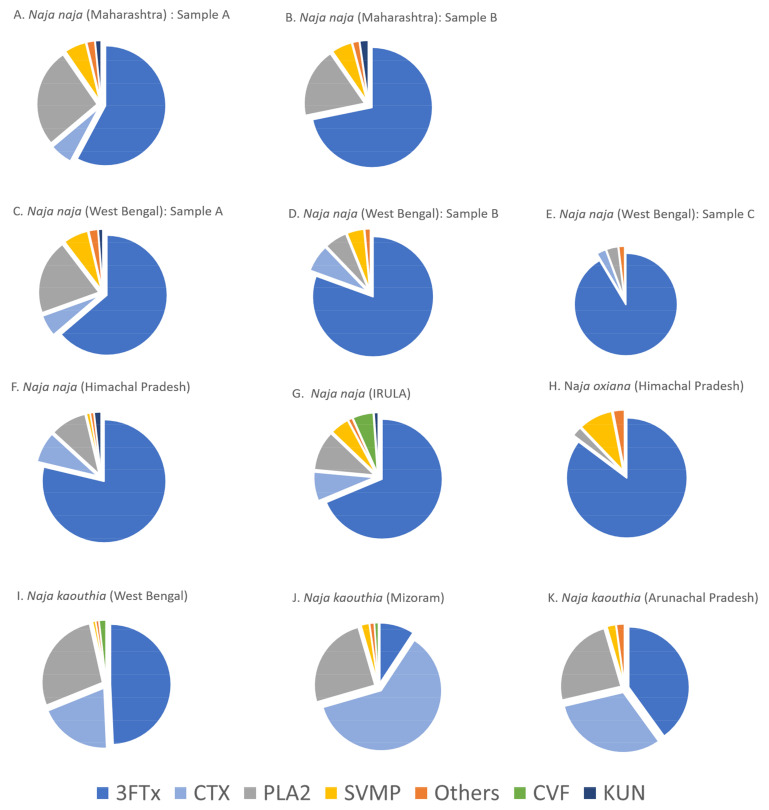
Pie charts highlighting the relative abundances of venom protein families in three different *Naja* species. Protein families: 3FTx, three-finger toxin; PLA_2_, phospholipase A_2_; CTx, cardiotoxins; KUN, Kunitz-type serine proteinase inhibitor-like protein; CVF, Cobra venom factor; SVMP, snake venom metalloproteinase. Toxins with relative abundances >1% were categorized into “Others” and include cysteine-rich secretory proteins (CRISP), l-amino acid oxidases (LAAO), nerve-growth factors (NGF), venom factors, nucleotidases, and phosphodiesterases (PDE), acetylcholinesterase, serine proteases, and tissue plasminogens.

**Figure 5 toxins-15-00258-f005:**
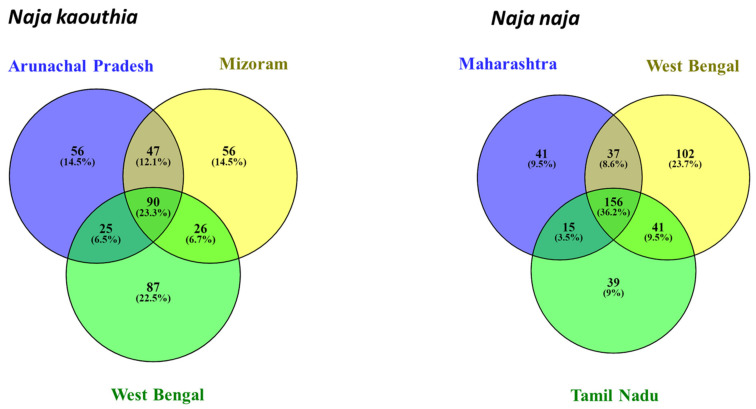
Venn diagrams representing the number of shared peptides (overlap) of individuals of the same species from different locations were constructed using Venny 2.1 [39]. Peptides were given unique IDs for their sequences and type of modifications from MS/MS spectra.

**Figure 6 toxins-15-00258-f006:**
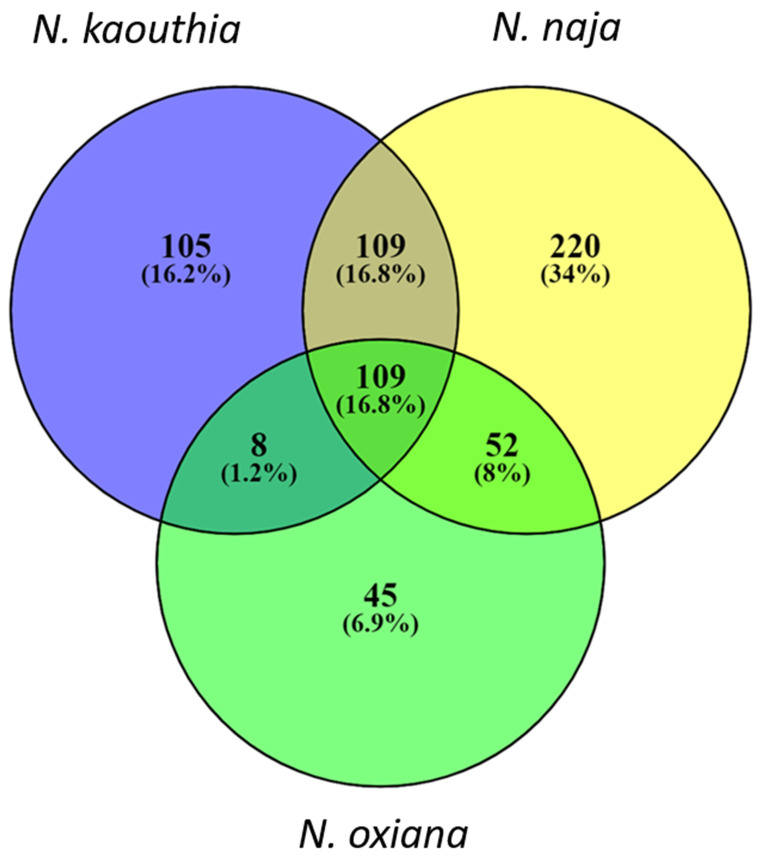
A Venn diagram representing the number of shared peptides in different *Naja* species was constructed using Venny 2.1 [39]. Peptides were given unique IDs for their sequences and type of modifications from MS/MS spectra.

**Figure 7 toxins-15-00258-f007:**
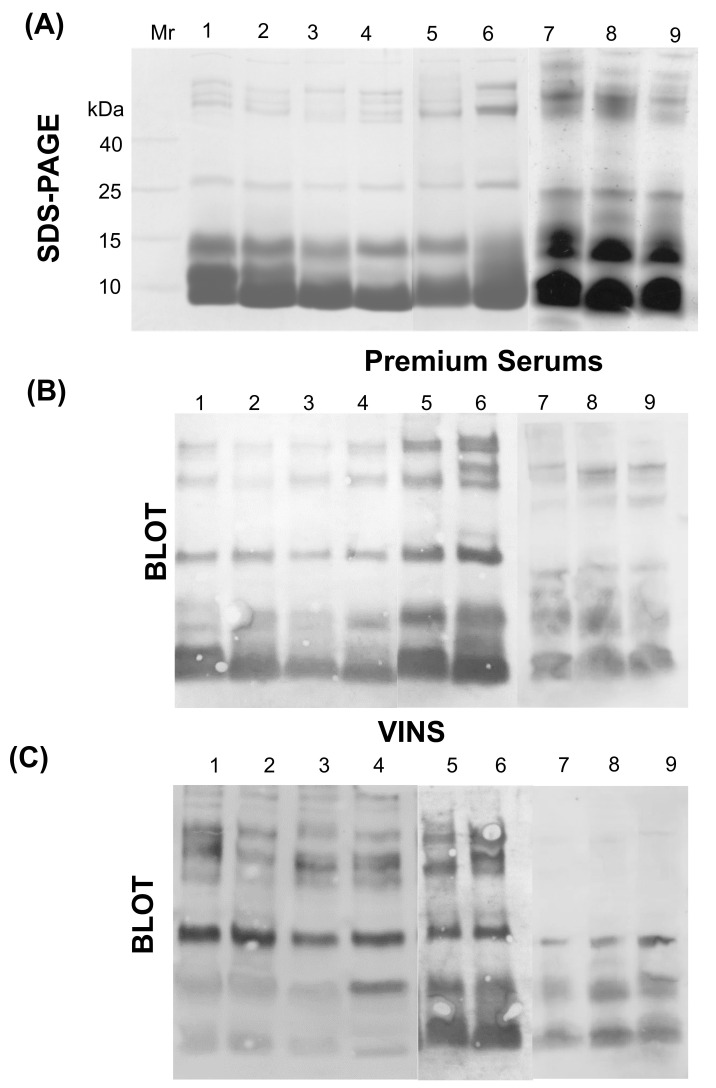
Immunoreactivity profile of polyvalent (PAV) antivenoms towards *Naja* venoms. (**A**) Reduced venom samples were applied to 10% Tris-Tricine SDS-PAGE. Venom proteins were then transferred to a PVDF membrane and incubated with different antivenoms—(**B**) Premium Serums PAV and (**C**) VINS PAV. Lane 1–4: *N. kaouthia* venom samples from Assam, Arunachal Pradesh, Mizoram, and West Bengal, respectively. Lane 5: *N. naja* venom from the Irula Co-op; Lane 6: *N. oxiana* venom from Himachal Pradesh; Lane 7–9: *N. naja* venom samples from Himachal Pradesh, Maharashtra, and West Bengal respectively.

**Figure 8 toxins-15-00258-f008:**
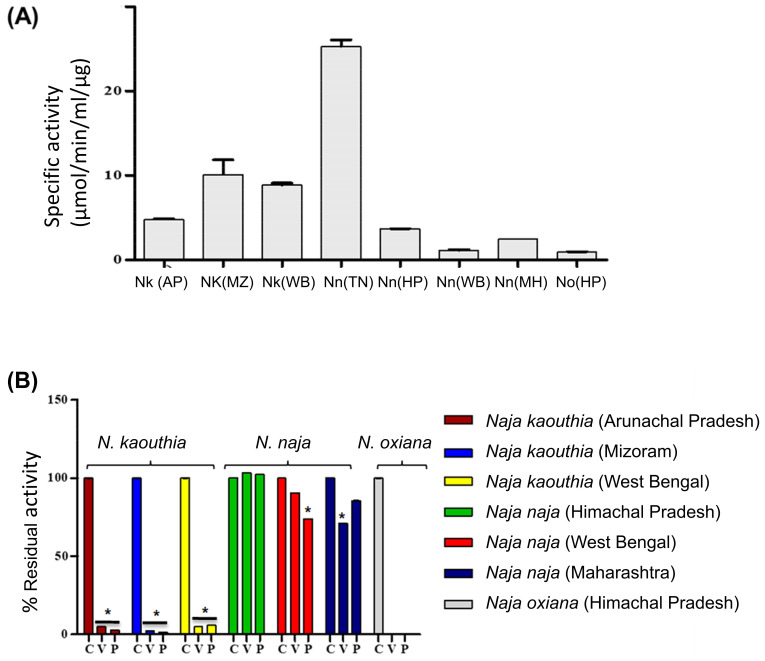
(**A**) Venom PLA_2_ activities of *Naja* samples. The amount of substrate hydrolysis was quantified at 414 nm using a spectrophotometer. (**B**) Neutralization of venom PLA_2_ activity. Venom was incubated with two different polyvalent antivenoms (V: VINS; P: Premium Serums) at a venom: antivenom ratio of 1:100 (protein/protein). The percentage residual activity was calculated considering the activity of the respective crude venoms without antivenom as 100%. C indicates venom PLA_2_ activity in absence of antivenom. * indicates *p* value < 0.005.

**Figure 9 toxins-15-00258-f009:**
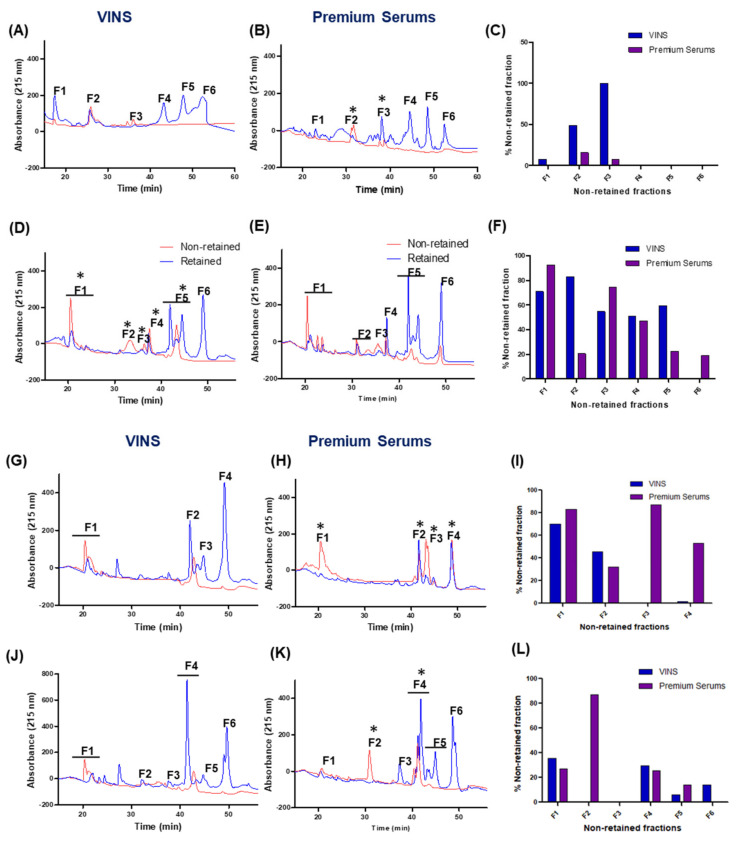
Immunorecognition profiles of VINS and Premium Serums polyvalent antivenoms using *N. oxiana* venom from Himachal Pradesh (**A**,**B**), and *N. kaouthia* venoms from Arunachal Pradesh (**D**,**E**), Mizoram (**G**,**H**), West Bengal (**J**,**K**). Panel (**C**,**F**,**I**,**L**) display a comparison of percentage of proteins remaining unbound to the two affinity columns. * indicates peaks subjected to proteomic identification by mass spectrometry.

**Table 1 toxins-15-00258-t001:** Details of venom collection with location, species, and sample ID.

Sample ID	Species	Location
15.47	*Naja kaouthia*	Papum Pare District, Arunachal Pradesh
16.24	*Naja kaouthia*	Aizawl District, Mizoram
17.25	*Naja kaouthia*	Hooghly District, West Bengal
18.41 (Sample A)	*Naja naja*	Kangra District, Himachal Pradesh
18.43 (Sample B)	*Naja naja*	Kangra District, Himachal Pradesh
15.35 (Sample A)	*Naja naja*	Ahmednagar District, Maharashtra
15.37 (Sample B)	*Naja naja*	Ahmednagar District, Maharashtra
15.73 (Sample A)	*Naja naja*	Hooghly District, West Bengal
15.74 (Sample B)	*Naja naja*	Hooghly District, West Bengal
17.24 (Sample C)	*Naja naja*	Hooghly District, West Bengal
-	*Naja naja*	Irula Snake Catcher’s Cooperative, Tamil Nadu
17.v18	*Naja oxiana*	Chamba District, Himachal Pradesh

**Table 2 toxins-15-00258-t002:** Characteristics of the Indian polyvalent antivenoms used in this study.

Manufacturer	Batch No. & Expiry	Active Substance	Protein per Vial	Neutralization Potency per Vial (mg venom/mL Antivenom)
Premium Serums and Vaccines Pvt. Limited	212013; 08/2023	F(ab) 2	629.95 mg	*Naja naja* venom (0.6 mg), *Daboia russelii* venom (0.6 mg), *Bungarus caeruleus* (0.45 mg), *Echis carinatus* (0.45 mg)
VINS Bioproducts Limited	01AS15007; 01/2019	F(ab) 2	487.05 mg	*Naja naja* venom (0.6 mg), *Daboia russelii* venom (0.6 mg), *Bungarus caeruleus* (0.45 mg), *Echis carinatus* (0.45 mg)

## Data Availability

All the proteome data generated using mass spectrometry in this study have been deposited in the ProteomeXchange Consortium via the PRIDE database under the accession number [PXD027361]. The data can be accessed using the Username reviewer_pxd027361@ebi.ac.uk and Password jyLMExaR.

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
