# Peer review of "Multilevel Comparison of Indian Naja Venoms and Their Cross-Reactivity with Indian Polyvalent Antivenoms"

_toxins, 2023, doi:10.3390/toxins15040258_

Round 1

Reviewer 1 Report

The manuscript fits with the journal’s scope. The aim of the study was to assess the venom composition across Naja species found in mainland India, as well as their cross-reactivity with two available polyvalent antivenoms. I find the article very informative and useful since the results might serve as a starting point for guidance to manufacturers when choosing the venoms that should be included in the immunization mixture in order to improve the effectiveness of their products. So, scientific significance of the investigation is highly relevant. The objectives of the study are well stated and clearly articulated, and its design is appropriate. The methods are described detailly. Results are presented in easy-to-follow manner. Conclusions are supported by the data and provide the foundation for advancement of snakebite management. I recognize only one limitation of the investigation – in vivo lethal toxicity neutralization assay, the gold standard for the evaluation of the antivenom’s applicability in the treatment of envenoming, is missing although the results with different venom/antivenom combinations would significantly contribute to the value of the manuscript. Maybe the authors could comment it briefly as an option for further study?

Definitively, before the paper acceptance, some steps concerning the clarity should be undertaken. I do feel that the improvement of language could solve the problem. Grammar and spelling check are highly recommended.

Also, some minor revision should be performed. Some typos that caught my eye are listed below.

I suggest “antivenoms” instead of “antivenom” in article title.

The scientific names of species should be italicized (for example, Naja in figure captions).

The space between number value and unit should be added throughout the manuscript.

“2” in “PLA2” should be in subscript.

When mentioning CTx in plural, “s” should be added (CTxs). Also, for cardiotoxin the authors sometimes use abbreviation CTX, sometimes CTx. Consistency in nomenclature should be assured in general.

In line 264 the authors should specify the antivenom they are writing about. Both maybe?

In line 550 I suggest replacement of “double and triple charged…” with “doubly and triply charged…”.

In few places the state of reagent manufacturer is missing (for example, in lines 574 and 615).

In line 603 “(g/g)” should be changed to “(m/m)”.

Author Response

The reviewer comments are in italics, followed point-by-point by our response in plain type.

The manuscript fits with the journal’s scope. The aim of the study was to assess the venom composition across Naja species found in mainland India, as well as their cross-reactivity with two available polyvalent antivenoms. I find the article very informative and useful since the results might serve as a starting point for guidance to manufacturers when choosing the venoms that should be included in the immunization mixture in order to improve the effectiveness of their products. So, scientific significance of the investigation is highly relevant. The objectives of the study are well stated and clearly articulated, and its design is appropriate. The methods are described detailly. Results are presented in easy-to-follow manner. Conclusions are supported by the data and provide the foundation for advancement of snakebite management.

We thank the reviewer for their favourable comments.

I recognize only one limitation of the investigation – in vivo lethal toxicity neutralization assay, the gold standard for the evaluation of the antivenom’s applicability in the treatment of envenoming, is missing although the results with different venom/antivenom combinations would significantly contribute to the value of the manuscript. Maybe the authors could comment it briefly as an option for further study?

We recognise that in-vivo assays are still the gold standard for antivenom evaluation, and indeed this is already acknowledged in the manuscript (“In-vivo venom neutralization assays with antivenom remains the standard test for determining antivenom efficacy”) but they are expensive and ethically problematic and we feel that the results in this manuscript are still a valuable contribution without them. Nevertheless, we have added to the sentence already in the discussion as requested, which now reads: “In-vivo venom neutralization assays with antivenom remains the standard test for determining antivenom efficacy. However, prior investigations using in-vitro assays to estimate neutralization efficacy of antivenoms are recommended to reduce the use of mice and the data presented here can be used in further in-vivo studies”.

Definitively, before the paper acceptance, some steps concerning the clarity should be undertaken. I do feel that the improvement of language could solve the problem. Grammar and spelling check are highly recommended.

Grammar and spell-checking has been done by the corresponding author, who is a native English speaker.

Also, some minor revision should be performed. Some typos that caught my eye are listed below.

I suggest “antivenoms” instead of “antivenom” in article title.

Done

The scientific names of species should be italicized (for example, Naja in figure captions).

The use of italics for scientific names serves to differentiate these from plain text. Hence, when other text is in italics, it is appropriate to use plain text to contrast with this. However, there were exceptions in the legends to Figure 1, 2, 7 and 8, and in the figure headings in Figures 2, 4 and 6, and these have been corrected.

The space between number value and unit should be added throughout the manuscript.

Checked and added where appropriate.

“2” in “PLA2” should be in subscript.

Checked and changed where appropriate (one sentence found).

When mentioning CTx in plural, “s” should be added (CTxs). Also, for cardiotoxin the authors sometimes use abbreviation CTX, sometimes CTx. Consistency in nomenclature should be assured in general.

Checked and changed where necessary to CTx or CTxs.

In line 264 the authors should specify the antivenom they are writing about. Both maybe?

The line numbers used by this reviewer did not correspond to those on the downloaded ms. However, we believe this is in reference to this sentence: “In contrast, PLA2 activities of N. naja venoms were not effectively neutralized by antivenom despite being the species included in the immunizing mixture, particularly in the case of N. naja from Himachal Pradesh”. We have added the word “either” before antivenom here.

In line 550 I suggest replacement of “double and triple charged…” with “doubly and triply charged…”.

Changed

In few places the state of reagent manufacturer is missing (for example, in lines 574 and 615).

It is not clear what the reviewer is referring to here as the lines mentioned do not refer to any materials in our version of the ms. We have added further details in sections 4.2, 4.4, 4.5 and 4.6.

In line 603 “(g/g)” should be changed to “(m/m)”.

This was on line 620 in our version of the ms. Changed.

Reviewer 2 Report

The ms entitled “Multilevel comparison of Indian Naja venoms and their cross-reactivity with Indian polyvalent antivenom” is focused and updated-informative on elapid snake research venomics from India. The authors of the present work aim to provide a formal categorization and quantification of the heterogeneity venoms at an individual level of the most representative elapid snake venoms from India and their PLA2 neutralization with antivenoms from the same country. So, the research brings interesting conclusions although some of them have been reported before such the lack of neutralization of some neurotoxins and PLA2s. Yet, it will be interesting to give to the readers a little bit of more information that is not perceptive within the field of snake venomics. For example, concerning the PLA2 activities of N. naja venoms, which were not effectively neutralized by antivenoms, a discussion concerning the differences of some PLA2 amino acid sequences compared among N. naja, N. kaouthia and N. oxiana could be interesting, or perhaps this lack of effectivity against N. naja PLA2s could be because of the different amount of PLA2s in these venoms? Please discuss; moreover, in the category of antivenoms, it will be important to include in the text the ED50 (this could be from literature or from the same vial specifications) of antivenoms from VINS Bioproducts and Premium Serums & Vaccines. 

Minor

Please include the same table format for tables S1 and S2, Toxin family, Coverage [%], MW [kDa], Calculated MW [kDa], Score Sequest HT, Percent abundance.

Why the protein concentrations of venoms were estimated by Lowry and the antivenoms were quantified using Nanodrop 2000? It could be better both by Lowry.

Author Response

The reviewer comments are given in italics, followed point-by-point by our response in plain type.

The ms entitled “Multilevel comparison of Indian Naja venoms and their cross-reactivity with Indian polyvalent antivenom” is focused and updated-informative on elapid snake research venomics from India. The authors of the present work aim to provide a formal categorization and quantification of the heterogeneity venoms at an individual level of the most representative elapid snake venoms from India and their PLA2 neutralization with antivenoms from the same country. So, the research brings interesting conclusions although some of them have been reported before such the lack of neutralization of some neurotoxins and PLA2s.

We thank the reviewer for their favourable comments.

Yet, it will be interesting to give to the readers a little bit of more information that is not perceptive within the field of snake venomics. For example, concerning the PLA2 activities of N. naja venoms, which were not effectively neutralized by antivenoms, a discussion concerning the differences of some PLA2 amino acid sequences compared among N. naja, N. kaouthia and N. oxiana could be interesting, or perhaps this lack of effectivity against N. naja PLA2s could be because of the different amount of PLA2s in these venoms? Please discuss.

We have added this to the Discussion:

“Although the presence of similar PLA2 isoforms were reported among the studied N. naja proteomes, significant variation in the percentage composition suggests differences in the expression of these shared PLA2 genes. Such quantitative differences could not only account for variable envenoming manifestations but may also be responsible for poor performance of polyvalent antivenoms in neutralization of the PLA2 activity of N. naja venoms. The proteomics data showed higher PLA2 abundance in N. kaouthia venoms, but fewer isoforms, compared to N. naja. Hence, the better neutralization of N. kaouthia could be because of lower isoform diversity in their venoms but also because the PLA2 isoforms from N. kaouthia venoms are also found in the N. naja venom from the Irula Co-op”.

Moreover, in the category of antivenoms, it will be important to include in the text the ED50 (this could be from literature or from the same vial specifications) of antivenoms from VINS Bioproducts and Premium Serums & Vaccines. 

The following text has been added to the Discussion:

“As earlier reported, Premium Serums antivenom (Batch No. 212013) displayed neutralization potency towards N. kaouthia venom from West Bengal (ED50 value, 0.156 mg/ml) while completely failing to neutralize the lethal effects of N. kaouthia venom from Arunachal Pradesh (Senji Laxmi et al. 2019). Also, VINS polyvalent antivenom was effective in neutralizing the lethal effects of N. kaouthia venom from Northeast India (ED50 value, 76.38 venom mg/antivenom g) (Deka et al. 2019). These studies perhaps indirectly support our observation that polyvalent antivenoms partially immunorecognized venom proteins of tested samples”.

Minor

Please include the same table format for tables S1 and S2, Toxin family, Coverage [%], MW [kDa], Calculated MW [kDa], Score Sequest HT, Percent abundance.

The formats for supplementary tables S1 and S2 has been changed as indicated above; however, we have not included protein abundance for table S2 as we haven't mentioned it in the manuscript.

Why the protein concentrations of venoms were estimated by Lowry and the antivenoms were quantified using Nanodrop 2000? It could be better both by Lowry.

In our study, the protein concentration of antivenoms were quantified using a Nanodrop 2000 (Thermo Fisher Scientific, USA) with the protein A280 method. The method does not need a standard curve and protein concentration in antivenoms is calculated using mass extinction of 13.7 L/gm-cm at 280nm for 10mg/ml IgG solution. The application of Nanodrop Protein A280 method is quick, requires low sample volume (2µl) and gives approximate concentration. However, protein measurement with the Nanodrop is not suitable for complex mixtures of proteins. Hence, we used Lowry’s method for protein estimation of venom samples, using Bovine Serum Albumin (BSA) for preparation of the standard curve.